# Macrophages in Atherosclerosis, First or Second Row Players?

**DOI:** 10.3390/biomedicines9091214

**Published:** 2021-09-13

**Authors:** Eloïse Checkouri, Valentin Blanchard, Olivier Meilhac

**Affiliations:** 1INSERM, UMR 1188 Diabète Athérothrombose Thérapies Réunion Océan Indien (DéTROI), Université de La Réunion, 97400 Sainte-Clotilde, France; eloise.checkouri@gmail.com (E.C.); valentin.blanchard@hli.ubc.ca (V.B.); 2Habemus Papam, Food Industry, 97470 Saint-Benoit, France; 3Departments of Medicine, Centre for Heart Lung Innovation, Providence Healthcare Research Institute, St Paul’s Hospital, University of British Columbia, Vancouver, BC V6T 1Z4, Canada; 4CHU de La Réunion, INSERM, CIC1410, 97500 Saint-Pierre, France

**Keywords:** macrophages, vascular smooth muscle cells, atherosclerosis, animal models, therapy

## Abstract

Macrophages represent a cell type that has been widely described in the context of atherosclerosis since the earliest studies in the 17th century. Their role has long been considered to be preponderant in the onset and aggravation of atherosclerosis, in particular by participating in the establishment of a chronic inflammatory state by the release of pro-inflammatory cytokines and by uncontrolled engorgement of lipids resulting in the formation of foam cells and later of the necrotic core. However, recent evidence from mouse models using an elegant technique of tracing vascular smooth muscle cells (VSMCs) during plaque development revealed that resident VSMCs display impressive plastic properties in response to an arterial injury, allowing them to switch into different cell types within the plaque, including mesenchymal-like cells, macrophage-like cells and osteochondrogenic-like cells. In this review, we oppose the arguments in favor or against the influence of macrophages versus VSMCs in all stages of atherosclerosis including pre-atherosclerosis, formation of lipid-rich foam cells, development of the necrotic core and the fibrous cap as well as calcification and rupture of the plaque. We also analyze the relevance of animal models for the investigation of the pathophysiological mechanisms of atherosclerosis in humans, and discuss potential therapeutic strategies targeting either VSMCs or macrophage to prevent the development of cardiovascular events. Overall, although major findings have been made from animal models, efforts are still needed to better understand and therefore prevent the development of atherosclerotic plaques in humans.

## 1. Introduction

Cardiovascular diseases represent about 31% of total deaths worldwide, constituting the first cause of total mortality [1]. Atherosclerosis, the major cause of cardiovascular diseases, is an arterial pathology characterized by the chronic accumulation of lipid components within the arterial wall, leading to its hardening and eventually to the complete obstruction of the blood flow [2].

First, endothelial stress and accumulation of cholesterol-rich lipoproteins containing apolipoprotein B (apoB) in the subendothelial space trigger an immune and inflammatory response, promoting the recruitment of monocytes-derived macrophages. Second, cholesterol-rich lipoproteins, particularly low-density lipoproteins (LDL), can undergo oxidative modifications in the intima where they can form aggregates. This leads to their uncontrolled uptake by macrophages and resident vascular smooth muscle cells (VSMCs) via scavenger receptors, resulting in the formation of lipid-laden cells called “foam cells”.

Due to their omnipresence during all stages of atherosclerosis, monocytes and macrophages are regarded as the main actors to this pathology. However, the central role of these cells in plaque formation and progression tends to be controversial since resident, undifferentiated VSMCs have recently been shown to be precursors of different cell types including foam cells in early, middle and late stages of atherosclerosis [3,4,5].

This review aims to evaluate and compare the respective roles of macrophages and VSMCs in atherosclerosis. In this work, we highlight the different arguments in favor of the involvement of each cell type at the different stages of atherosclerosis.

### 1.1. At the Beginning of Atherosclerosis

Everything starts in the circulatory system. In a context still not very conducive to atherosclerosis, a first stress occurs at the surface of the endothelium of arteries: the shear stress. This stress results from a set of mechanical phenomena (vortices, vortex) related to the blood flow at the curvatures of the arteries [6]. This “endothelial shear stress” is believed to cause the first disorganization of the endothelial cells. Despite their polygonal shape allowing them to adapt to a modified blood flow, endothelial cells located at the bifurcation zones are strongly impacted by mechanical stresses [7]. In addition, the absence of glycocalyx, a thick layer enriched in carbohydrates at the surface of endothelial cells, make them more exposed to circulating apoB100-containing lipoproteins [remnant chylomicron, very low-density lipoproteins (VLDL) and remnant VLDLs, IDLs, LDLs and Lp(a)] whose accumulation in the intima is exacerbated in hyperlipidemic conditions [8]. The mechanisms by which these lipoproteins enter the intima are unclear but several pathways have been described including the transcellular route and transcytosis [9,10].

The transcellular route involves structural changes in cell-to-cell junctions including tight and adherens junctions, allowing the passage of molecules larger than 6 nm in diameter. In pathological conditions such as dyslipidaemias, diabetes, hypertension, obesity and smoking, the endothelium becomes more permeable to atherogenic lipoproteins [9]. During transcytosis, lipoproteins are taken up by endothelial cells via direct transport by cell surface receptors and via indirect transport from the apical to the basal membrane involving caveolae [11]. In receptor mediated-transport, lipoproteins are endocytosed and recycled within endosomes, before being exocytosed to the opposite side of the endothelium. Scavenger receptor B1 (SR-B1) [12,13] and activin receptor-like kinase 1 (ALK1) [14] are the only two receptors that have been shown to mediate LDL transcytosis across the endothelium. The LDL receptor (LDLR) has also been a candidate for LDL transcytosis, especially across the blood-brain barrier [15], but recent evidence showed that degradation of the LDLR by its natural inhibitor proprotein convertase subtilisin/kexin type 9 (PCSK9) has no effect on LDL transcytosis [12].

Lipoprotein accumulation in the intima is not a random process in the vascular system. Compared with arteries resistant to atherosclerosis (e.g., ascending aorta and hepatic arteries), atherosclerosis-prone arteries (e.g., abdominal and descending aorta, coronary and carotid arteries) are characterized by diffuse thickened intima. Diffuse intimal thickening (DIT) is an early event present in human arteries in fetuses and infants before the development of atherosclerosis [16]. DIT is mainly composed of vascular smooth muscle cells (VSMCs) with abundant proteoglycans and elastic fibers but is devoid of lipids and macrophages. DIT contributes to the development of atherosclerosis by promoting the initial retention of lipoproteins via ionic interactions between negatively charged proteoglycans and positively charged apolipoproteins such as apoB100 and apoE [17,18]. While lipoproteins accumulate in DIT, medial VSMCs of mono- or oligoclonal origin also migrate to the intima where they proliferate and further contribute to DIT [19,20]. In DIT, VSMCs of medial origin lose their contractile and spindle-shaped phenotype and acquire a synthetic and proliferative phenotype characterized by the downregulation of differentiated VSMC markers and upregulation of genes encoding extracellular matrix (ECM) components such as proteoglycans. However, it is not clear whether the VSMCs present in DIT in fetuses and infants originate from the surrounding media. This creates optimal conditions for the retention of penetrating lipoproteins within the DIT where they accumulate and aggregate over time, promoting their modification by reactive oxygen species (ROS) and enzymes (lipases and myeloperoxidase) [18]. The formation of foam cells comprising mainly foam cells derived from VSMCs and to a lesser extent foam cells derived from macrophages constitutes the first stage of early atherosclerotic lesions called fatty streaks [4].

In addition, during stressful conditions (presence of oxidized LDLs, shear stress, reactive oxygen species), endothelial cells initiate the secretion of cytokines (TNF-a, IL-1β, IL-6, IL-8) [21,22], chemokines (CCL2,CX3CL1) [23,24,25] and express adhesion molecules (ICAM-1, VCAM-1) [26]. This primary defense reaction aims at promoting the recruitment of leukocytes, which plays an important role in the development and exacerbation of atherosclerosis.

### 1.2. Monocytes as Initiators of Atherosclerosis

The implication of monocytes has been described as a risk factor for cardiovascular diseases, notably for coronary heart disease [27]. Dyslipidemia (including hypercholesterolemia) is a risk factor for atherosclerosis, as it promotes LDL-C accumulation within the arterial wall [28]. Studies on animal models showed that hypercholesterolemia was associated with an increase in circulating monocytes and then would promote the aggravation of atherosclerosis [29]. Moreover, their presence in atherosclerotic lesions has been described in both humans and animal models [30,31,32]. Indeed, monocytes are reported to initiate the atherosclerotic process via different steps: rolling, adhesion, activation and migration through the activated endothelium.

Monocytes come from the differentiation and proliferation of hematopoietic stem and multipotential progenitor cells (HSPCs) from the bone marrow. Studies show that cellular environments associated with rheumatoid arthritis or hypercholesterolemia enhance HSPC differentiation into myeloid cells (monocytes and neutrophils) [33,34,35].

Monocytes can be categorized according to glycoproteins expressed at their cell membranes (Figure 1). Monocytes that will differentiate into macrophages express high levels of the glycoprotein Ly6C (lymphocyte antigen 6C “Ly6C^high^” in mice) known as CD14 in humans (CD14^++^CD16^−^) and will actively contribute to atherosclerosis in contrast to monocytes expressing low Ly6C levels and CD14^+/low^CD16^+^ for humans [36,37]; Ly6C^high^ and CD14^++^CD16^−^ are considered as precursors of M1 macrophages, producing inflammatory cytokines and leading to foam cell formation, whereas Ly6C^low^ and CD14^+/low^CD16^+^ were described as resident and patrolling cells, responding to infections, producing high levels of anti-inflammatory cytokines and being able to remove damaged cells from the vasculature [38,39,40]. Recent studies put in light the conversion of monocytes Ly6C^high^ into Ly6C^low^ [41,42,43], an important process that may lead to atherosclerosis regression [44].

The trafficking of monocytes to the damaged area of the endothelium is regulated by interactions between secreted chemokine (C-C motif) ligands (CCLs) and chemokine (C-C motif) receptors (CCRs) present at the cell surface. Each monocyte subset has a different phenotype: CX3CR1^low^ and CCR2^high^ are associated with inflammatory monocytes (Ly6C^high^; CD14^++^CD16^−^), whereas CX3CR1^high^ and CCR2^low^ are associated with resident monocytes [45]. It has been described that CXC3CR1 expression is stimulated by CCL-2 and leads to higher adhesion of monocytes [46]. On the other side, CCR2 determines the number of Ly6C^high^ cells and has an impact on the release of monocytes from the bone marrow, even if it is not required for the monocyte migration [47].

Different models of CCLs or CCRs deficient mice put in light the role of these proteins in the monocyte recruitment and the development of atherosclerosis. For example, CCR2^−/−^ mice showed a delay in clearance after an intramuscular injection of LDL-C, suggesting that CCR2 is essential for monocyte migration to the lesion [48]. CX3CR1^−/−^ apoE^−/−^ mouse model showed an impaired survival of Ly6C^low^ monocytes and enhanced atherosclerosis [49]. Furthermore, studies carried out after inhibition of CCL2, CX3CR1 and CCR5 in apoE^−/−^ mice highlighted the contribution of CCL2, CX3CR1 and CCR5 in the monocyte accumulation process and their impact on monocyte differentiation [50,51]. Another finding from Gu et al. showed that CCL2 depletion in LDLR^−/−^ mice was associated with reduced lipid depositions (less 83%) in the aortas [52].

Triggering receptor expressed on myeloid cells-like 4 (TREML4) is a positive regulator of Toll-like receptor (TRL) signaling which leads to the initiation of intracellular signaling pathways that elicit the expression of inflammatory genes. Recently, it has been shown that macrophages from TREML4^–/–^ mice were hyporesponsive to TLR7 agonists and failed to produce type I interferon, a major inducer of the M1 pro-inflammatory phenotype [53]. In addition, TREML4 is highly expressed in patients with atherosclerosis [54,55]. In line with these findings, TREML4 has recently been shown to regulate the expression of genes related to inflammation and lipid metabolism in M1 macrophages [56]. In this study, they also showed that combined apoE and TREML4-deficient mice developed less atherosclerosis with reduced macrophage and monocyte content as well as decreased collagen deposition, compared with apoE-deficient control mice [56]. Further research is needed to decipher the role of TREML4 in human atherosclerosis.

After reaching the damaged/activated endothelium, monocytes roll to the lesion by interacting with the P-selectin glycoprotein ligand-1 (PSGL-1) that can bind with P, E or L-selectins [57]. Unlike P and E-selectins which are expressed by activated endothelial and platelet, L-selectins are only expressed by leukocytes [57]. Additional interactions, in particular with endothelial adhesion molecules contribute to monocyte anchoring and transendothelial migration. Indeed, the integrin-type cell adhesion molecules, leukocyte function-associated antigen (LFA-1) and very late antigen (VLA-4) respectively bind to intracellular adhesion molecule (ICAM-1,2,-3) and vascular cell adhesion molecule (VCAM-1) or fibronectin [58,59]. Recent in vitro studies showed the implication of α-2,6 sialyltransferase 1 (ST6GAL1) in atherosclerosis related transendothelial migration [60].

In brief, in the atherosclerosis context, high blood levels of LDL-C and cytokines produced by the endothelium stimulate the production of two main subsets of monocytes from the bone marrow. One subset is known to be an atherosclerosis-aggravating phenotype of macrophages. Monocytes, following chemokine signaling roll, adhere and pass through the activated endothelium.

At the early stage of atherosclerosis, monocytes are recruited to the subendothelial space while SMCs remain wisely in the media layer. It would be rational to think that these macrophage precursors, the first that arrived on the scene, take the center stage in atherosclerosis.

## 2. Vascular Smooth Muscle Cells and Early Atherosclerosis

VSMCs form the medial layer of arteries and are responsible for both vascular tone and contractility. They express a range of proteins required for their contractile functions including alpha-smooth muscle actin (αSMA), smooth muscle myosin heavy chain (SMMHC or myosin 11 (MYH11)) and transgelin (TAGLN) [61,62]. Medial VSMCs also secrete extracellular matrix (ECM) proteins such as elastin, collagens and proteoglycans, which can modulate the elastic properties of large arteries [4].

While medial VSMCs are elongated and spindle-shaped, VSMCs in diffuse intimal thickening (DIT) are less elongated, display a cobblestone morphology and overexpress ECM protein-expressing genes [63]. DIT VMSCs have more synthetic organelles such as rough endoplasmic reticulum and ribosomes [64]. This is consistent with the fact that DIT is abundantly composed of VSMC-derived secretory products such elastin and proteoglycans which trap lipoproteins in the subendothelial space [18]. In contrast, DIT lacks macrophages [65,66]. As the first cells facing the arrival of blood lipids, synthetic DIT VSMCs undergo lipoprotein uptake leading to the formation of VSMC-derived foam cells. In vitro studies showed that synthetic VSMCs were more efficient at binding and accumulating lipoprotein cholesterol than contractile VSMCs, resulting in a higher propensity to become foam cells [4,67,68].

Owing to chronic lipid deposition into the arterial wall, DIT may progress to an early atheromatous lesion [4]. At this stage, several cell types including endothelial cells, platelets and inflammatory cells release proinflammatory mediators, leading to the dedifferentiation of medial VSMCs into modulated (also called synthetic) VSMCs which migrate from the media to the intima. This stage also known as pathological intimal thickening (PIT) corresponds to the fibroatheroma and progression of atherosclerosis towards complicated lesions. According to the original “response-to-injury” hypothesis of atherogenesis, this migration of phenotypically switched VSMCs occurs in part as a wound healing mechanism by overexpression of ECM components including proteoglycans [69]. However, many studies also highlighted the role of VSMC-derived cells in plaque growth and rupture.

VSMC phenotypic switching is modulated by transcription factors including myocardin (MYOCD) and its cofactor serum response factor (SRF), the two main mediators driving the SMC contractile phenotype [67,68] and Krüppel-like factor (KLF4), which is not expressed in contractile VSMCs. Wang et al. showed that together, MYOCD and SRF form a ternary complex over the CArG-box elements located on the promoter region of genes coding for contraction-related protein [67]. In contrast, KLF4 promotes phenotypic transition of VSMCs by inhibiting MYOCD/SRF interaction in response to mediators such as platelet-derived growth factor PDGF BB [70], oxidized phospholipids [71], cholesterol loading [72] and interleukin (IL) 1-β [73]. Depending on the presence or absence of these various mediators in DIT, modulated VSMCs may redifferentiate into multiple phenotypes such as macrophage-like, adipocyte-like, endothelial-like, osteochondrogenic-like or mesenchymal-like cells [3,74,75,76].

VSMCs can also undergo a phenotypic switch in response to environmental changes related to cell-cell and cell-ECM interactions [77]. For instance, in healthy arteries, medial VMSCs are surrounded by a basement membrane enriched in type IV collagen and laminin, which is in turn embedded in the interstitial matrix containing type I and III collagens. This complex network of proteins stabilizes the structure and function of the arterial wall, in part by sequestering contractile VSMCs within the media [78,79]. In response to arterial injury, increased activity of proteases such as matrix metalloproteinases (MMPs) leads to proteolytic degradation of the basement membrane and interstitial matrix [80]. Loss of contact between medial VSMCs and surrounding ECM induces dramatic phenotypic changes in VSMCs, which migrate toward the lesion in the intima. VSMCs also acquire a synthetic phenotype with high production of neo-synthesized ECM components such as proteoglycans, which contribute to lipid sequestration and retention in the intima [81].

Although macrophages are thought to play the role of initiators of fatty streaks, VSMCs are naturally present in both medial and intimal layers (at least in humans). Moreover, as major cells of DIT, VSMCs and their secretory products involved in early lipid retention seem to be crucial for the onset of atherosclerosis.

### 2.1. At the Origin of Foam Cell Formation: Macrophages or VSMCs?

Atherosclerotic lesion is an environment characterized by oxidative stress and inflammation. After exposure to subendothelial ox-LDL, monocytes undergo proliferation and differentiation into macrophages influenced by environmental factors, including the granulocyte-macrophage colony-stimulating factor (GM-CSF) and the chemokine CXC ligand 4 (CXCL4) [82,83,84].

Depending on environmental conditions (presence of cytokines, lipopolysaccharides (LPS), oxidized lipids...), monocyte differentiation may result in different macrophage phenotypes. For example, the better-known macrophage phenotypes are M1 (induced by T-helper 1/Th1 cytokine signature) and M2 (induced by T-helper 2/Th2 cytokine signature), which are respectively involved in atherosclerosis progression and resolution. Different subsets of M2 macrophages have been described (M2a–d) as well as other macrophage phenotypes such as Mox, Mhem and M4 [85]. After their migration into the intimal space, monocytes are rapidly confronted with various microenvironments, with different available metabolites that will direct their differentiation into M1 or M2 macrophages. Glucose, fatty acid and amino acid metabolism are thought to modulate macrophage phenotype and function at the different stages of atherosclerosis (for review, see [86]). For example, specific glucose fluxes have been shown to be utilized in M1- and M2-type macrophages in glycolysis and the pentose phosphate pathway [87]. This may have an impact on imaging of the atherosclerotic plaque using 18F-FDG PET, which relies on glucose uptake. It was shown that only LPS-stimulated macrophages, but not M1 or M2 polarized macrophages, exhibited an increased glucose uptake and could therefore be more easily detected by 18F-FDG PET [88]. Some less known atheroprotective phenotypes (HA-mac; M(Hb); Mhem) carrying out hemoglobin, cholesterol and erythrocyte clearance have also been described [89]. M1 and M2 macrophages have been described in both humans and mice: the M1 phenotype, sensitive to interferon; TNF- and LPS, which are known to secrete large amounts of pro-inflammatory cytokines; and the M2 phenotype, influenced by the cytokines IL-4 and IL-13, are more associated with the anti-inflammatory reaction [90,91]. The term “polarization” is commonly used in reference to their ability to switch from one phenotype to the other [90]. For instance, M2 macrophage polarization was described as mediated by the Stat6 pathway [92]. Ox-LDLs present in atherosclerotic lesions also promote monocyte differentiation into pro-inflammatory macrophages involving PPARγ expression [93]. A better understanding of the metabolic regulation of macrophages associated with their various phenotypes would provide better therapeutic and diagnostic targets.

An important point to note in the aggravation of atherosclerosis is linked to a massive secretion of cytokines leading to the dysfunction and death of the surrounding cells. At the origin of this phenomenon are the so-called “pro-inflammatory” macrophages. Depending on their level of differentiation, macrophages could aggravate or delay the progression of atherosclerosis. But what about VSMCs?

### 2.2. VSMC Foam Cell Formation

Cellular plasticity is an important feature of the atherogenic process, in which arterial resident cells need to adapt to changes in the environment. In particular, VSMCs are able to change phenotype and convert themselves into ECM synthetic cells, but also into macrophage-like cells in order to phagocyte the excess of lipids or cellular debris [94]. Several important studies have shown that upon switching to a synthetic phenotype, VSMCs progressively lose the expression of contractile proteins such as αSMA, MYH11 and TAGLN and acquire macrophages and leukocytes specific markers such as CD68, CD11b, F4/80, HLA-DR and LGALS3/Mac2 [72,95,96,97,98,99,100]. As a result, the contribution of VSMC-derived cells to all plaque cells based on immunohistochemical studies of atherosclerotic plaque sections has been underestimated.

Recently, the development of an elegant methodology for VSMC lineage tracing in animal models has allowed a precise study of the role and fate of VSMCs throughout the different stages of atherosclerosis as well as different regions of the lesions [97,98,99]. The principle of VSMC lineage tracing in atherosclerosis is to track vascular VSMCs and their progeny to study their behavior and migration during atherogenesis. This methodology relies on the conditional expression of reporter genes (e.g., GFP) exclusively in a specific cell line (e.g., VSMCs), allowing identification of both parents and progeny cells. In the case of VSMC tracing studies, Cre recombinase expression is under control of the promoter of a gene, which is exclusively expressed in the VSMC lineage, such as MYH11. Consequently, Cre recombinase is expressed in the same way as MYH11 and then recognizes LoxP sites flanking a stop sequence located at a locus carrying the reporter gene. Once the stop sequence is irreversibly removed from the parent cell DNA by the Cre recombinase, the reporter gene can be expressed uninterrupted and independently of MYH11 expression. The transmission of this gene to each descendant cell allows identification of all VSMC-derived cells by the signal resulting from the expression of the reporter gene (e.g., green fluorescence for GFP). In addition, Cre recombinase expression can be controlled by an inducible system permitting its activation at a given time upon the administration of an inducer such as tamoxifen. These conditional and inducible Cre recombinase systems allow the study of VSMC fate independently of the expression of marker genes such as MYH11 or αSMA, which are often downregulated in plaques [101].

Due to the expression of macrophage-specific markers by some VSMCs, the contribution of VSMC-derived foam cells to all plaque foam cells has long been underestimated or even ignored. VSMCs are now known to be the major cellular site of lipid accumulation, being the driving force of atherosclerosis progression. The first piece of evidence for this is that VSMCs, unlike macrophages, contribute to initial lipid retention in the intima of atherosclerosis-prone arteries in utero as well as in infants and children [16,69,102]. VSMCs also contribute significantly to the foam cell population in advanced plaques of adults [94]. A major study from the Francis group revealed that in human coronary artery sections, at least 50% of total intimal foam cells were derived from VSMCs and that about 40% of CD68+ cells were of VSMC rather than myeloid origin [103]. They also showed that the gene expressions of the cholesterol exporter protein ATP-Binding Cassette A1 (ABCA1) and lysosomal acid lipase (LAL), a cytosolic enzyme which hydrolyzes cholesteryl esters (the main form of cholesterol storage) to free cholesterol, were significantly reduced in VSMC-foam cells but not in macrophages-foam cells, suggesting that VSMCs may contain a much larger burden of lipids than macrophages, as they are unable to release excess lipid via ABCA1 and LAL [103,104]. In line with this observation, a study found that TNF ligand-related molecule 1A (TL1A), a vascular endothelial growth inhibitor reducing neovascularization, was able to inhibit the development of atherosclerosis by regulating VSMC (but not macrophage) foam cell formation by activating ABCA1, ABCG1 and cholesterol efflux by a liver X Receptor (LXR)-dependent signaling pathway [105].

Several studies reported mechanisms and processes by which VSMCs may accelerate foam cell accumulation in atherosclerosis. First, the secretory properties of intimal synthetic VSMCs contribute to ECM secretion by an increased proteoglycan production [102]. ApoB100-containing lipoproteins including LDL particles are retained and modified within the proteoglycan network built by VSMCs via hydrophobic interactions [106,107]. In vitro studies showed that VSMC foam cell formation can occur irrespective of oxidative [108] or enzymatic [109] LDL modifications. However, enzymatic modifications of LDL appear to be a more potent stimulus than oxidative modifications in terms of VSMC foam cell formation [110].

Unlike the early stage of DIT during which VSMCs are the only lipid-laden cells of intima, later stage lesions contain macrophage foam cells derived from activated monocytes [94]. In vitro study from Vijayagopal et al. focusing on VSMC-macrophage co cultures showed that direct cell-cell contact significantly increased both cholesterol accumulation and synthesis in VSMCs [111]. A more recent report from Weinert et al. demonstrated that this cell-cell interaction may in part involve lysosomal transfer of cholesterol from macrophages into VSMCs [112], which further increases their phagocytic activity. Interestingly, Niu et al. also showed that extracellular vesicles released by macrophage foam cells could promote VSMC migration and adhesion via ERK- and Akt-mediated phosphorylation [113]. However, the authors did not investigate whether such extracellular vesicles were also able to promote VSMC foam cell formation. In contrast, an earlier study reported that VSMCs were able to phagocyte cholesterol ester-rich droplets isolated from macrophage foam cells in a metabolic activity-dependent manner [114]. A more recent study confirmed that macrophages were able to transfer the surplus of cholesterol towards adjacent VSMCs, even in the absence of high-density lipoproteins or ABCA1 at the cell surface of macrophages [115].

All recent investigations, including lineage-tracing studies that can accurately estimate the number and the origin of each cell type in plaques, agree that VSMCs are the principal contributors to form foam cells at all stages of atherosclerosis.

### 2.3. Scavenger Receptors Expression, a Step towards Foam Cell Formation

The presence of foam cells is described at the early stages of atherosclerosis [116]. These cells are the result of an excessive and unregulated engorgement of modified cholesterol-rich lipoproteins (including ox-LDLs) by VSMC and macrophages [117]. It has been shown that ox-LDLs also trigger surface expression of “scavenger receptors” such as scavenger receptor A (SRA), lectin-like oxidized LDL receptor-1 (LOX-1) and cluster of differentiation (CD) 36 (CD 146 and CD 68 for humans), leading to foam cell formation [118,119,120,121]. In vitro studies also highlight the close link between TLR pathway and foam cell formation, notably supported by the increased lipid accumulation by macrophages after stimulation with bacterial lipopolysaccharides (LPS) [122,123,124]. While LDLR expression is downregulated in response to elevated intracellular cholesterol levels, scavenger receptor expression is unaffected by this feedback regulation, resulting in uncontrolled uptake and accumulation of cholesterol in VSMC and macrophages [125]. Nonetheless, scavenger receptors do not only play a role in oxidized lipoprotein uptake, but also act as lipid sensors [126].

The main scavenger receptors involved in atherosclerosis are presented in Table 1.

Ox-LDL uptake is followed by an enzymatic degradation in the late endosome/lysosome to release free cholesterol and fatty acids into the cytosol [148]. Free cholesterol can be reconditioned in cholesteryl esters by the mitochondrial acylCoA cholesterol acyl-transferase-1 (ACAT-1) and stored in the endoplasmic reticulum. In homeostatic conditions, cholesterol efflux may occur via neutral cholesteryl ester hydrolase (NCEH) activity and free cholesterol transfer to ABCA1, ABCG1 and SR-B1. However, the atherosclerotic context is an excess of lipids and an increased scavenger receptor expression at the expense of cholesterol efflux mediators [149,150]. This unregulated lipid uptake and their partial degradation in the lysosomes, leads to endoplasmic reticulum (ER) stress mediated by unfolded protein response (UPR) and activation of “cholesterol-induced apoptosis” (involving caspases, Jun-N-terminal stress activated protein kinases (JNK) and the transcription factor CHOP) [151,152,153].

Importantly, red blood cells also constitute another substrate for foam cell formation. Intraplaque hemorrhage is not only observed in the late stages of atherosclerotic lesions [154]. Erythrocyte extravasation may represent an important source of cholesterol, as their membranes are particularly rich in lipids including phospholipids and free, unesterified cholesterol. Indeed, glycophorin from the erythrocyte membrane is recognized by macrophage scavenger receptors. Beyond that, traces of erythrocytes at different stages of atherosclerosis (intimal thickening, fibrous cap atheroma, neovascularization, intraplaque hemorrhages) support their implication in promoting plaque initiation, progression, instability and rupture [155,156]. As a result, it is possible that the cholesterol responsible in foam cell formation may also come from damaged erythrocyte phagocytosis by macrophages [157,158].

The conclusion seems undeniable: macrophages already present in large numbers in the fatty streak are in charge of lipid plaque clearance. Unfortunately, this unregulated uptake of cholesterol that is only partially degraded in the lysosomes leads to foam cell formation, a well-known driver of atherosclerosis.

Macrophages are not the only cell type expressing scavenger receptors. Indeed, VSMCs are also able to form foam cells.

CD36 is a major receptor for ox-LDL uptake by macrophages. Matsumoto et al. reported that certain but not all primary human aortic VSMCs were able to express CD36, suggesting a clonal difference between them [159]. During the same year, Ricciarelli et al. indicated that Vitamin E downregulated CD36 expression by reducing its promoter activity and led to a parallel reduction of ox-LDL uptake by human aortic VSMCs [160]. More recently, Xue et al. reported that hyperglycaemia-induced VSMC foam cell formation was mediated by a CD36-dependent ox-LDL uptake and was associated with increased oxidative stress and NF-κB pathway activation [161]. In addition to its LDL uptake capacity, CD36 is also able to induce VSMC foam cell formation via the uptake of free fatty acids such as oleic acid [144]. CD36 is likely the most important scavenger receptor in VSMCs, as antisense oligonucleotides directed against CD36 mRNA are capable of inhibiting the up-take of ox-LDL by up to 80% [143].

VSMC foam cells display several types of scavenger receptors involved in excess lipid loading. Like macrophages, SR-AI is expressed by VSMCs of both rabbit [144] and human [162] atherosclerotic lesions. In human VSMCs, EGF and TGF-beta 1 stimulate SR-AI activity in the presence of IGF 1 or PDGF BB [163]. Pitas’ group showed that SR-AI mRNA expression in VSMCs was upregulated by redox-sensitive transcription factors such as NFB activating protein-1, c-Jun and CCAAT enhancer-binding protein beta in the presence of ox-LDL and ROS was correlated with cyclooxygenase-2 expression and calcium flux [147,164,165]. Although SR-AI is a key receptor for the uptake of LDLs by macrophages, its involvement in VSMC cholesterol burden appears to be negligible [166].

LOX-1 is present on VSMC membranes and its expression is increased in human atherosclerotic lesions in response to ox-LDL and proinflammatory cytokines such as IL-1α, IL-1β and TNF-α [167]. Hypertension and high blood cholesterol also induced LOX-1 expression in VSMCs [168,169]. Mukai et al. reported that heparin-binding EGF (HB-EGF) significantly increased LOX-1 expression and Ox-LDL uptake by promoting the phosphorylation of ERK, p38 MAPK and Akt in VSMCs [170].

The scavenger receptor for phosphatidylserines and ox-LDL (SR-PSOX) is a transmembrane chemokine also known as CXCL16 present on human aortic VSMCs cell sur-face [164]. SR-PSOX levels at the cell surface of VSMCs are upregulated by IFN-γ and followed by an intracellular accumulation of ox-LDL [165]. Although SR-PSOX has been identified as being responsible for ox-LDL uptake by VSMCs, studies are lacking that decipher the role of this receptor in the formation of VSMC-derived foam cells.

Additional receptors such as receptors for advanced glycation end products (RAGE) [171,172], P2RY12 receptor [172] or LRP1 [173,174] have recently been suggested to be key mediators of VSMC-derived foam cell formation but further investigations are needed to clarify their precise role in atherosclerosis.

Image evidence indicates that the phenotypic change in VSMCs could be observed by immunohistology.

Our histological analyses on human carotid samples with atherosclerotic lesions show areas where VSMCs seem to express both αSMA and CD68, which are generally reputed to be expressed by macrophages. Our results presented in Figure 2 suggest a phenotypic change in VSMCs and their contribution to foam cell formation.

They are all about markers; whereas observations from the 1970s were based on ultra-structure, mainly using electron microscopy, with the advent of antibodies and immunological techniques, most evidence supporting the role of macrophages in atherosclerosis is currently based on “specific markers”. However, as previously discussed, VSMC plasticity may induce the expression of membrane markers thought for decades to be specific for macrophages.

### 2.4. Towards the Necrotic Core Formation

In a context of lipid accumulation, oxidative stress and high concentrations of cytokines secreted by all vascular and inflammatory cells, macrophages trigger apoptosis and potentially secondary necrosis due to the lack of an efficient phagocytosis process. Foam cells, damaged cells and debris constitute the necrotic core. The development of the necrotic core is also mediated by deficient phagocyte efferocytosis (clearance of apoptotic cells by phagocytosis); indeed, foam cells (macrophages or smooth muscle cells) have a decreased capacity to eliminate cell debris and apoptotic bodies [175]. Consequently, early lesions display more effective efferocytosis as compared to vulnerable plaques because of the lower number of apoptotic cells [176,177]. At this stage of atherosclerosis, the implication of macrophages seems undeniable, both due to their presence at the very initiation of atheroma and due to their contribution to local inflammation and necrotic core formation.

Unlike macrophages, VSMCs are not specialized in removing debris and apoptotic cells. Yet, several studies have shown that VSMCs also have phagocytic properties [178,179]. For example, Kolb and colleagues showed that senescent red blood cells are phagocytized by primary cultures of VSMCs in vitro. However, the phagocytosis capacity of smooth muscle cells seems to be dependent on the surrounding conditions. Fries and colleagues showed by electron microscopy that efferocytosis of apoptotic cells by VSMCs was enhanced by the release of cytokines such as CCL2, cytokine-induced neutrophil chemoattractant-1 and TGF-beta 1 by neighboring VSMCs [180]. In contrast, when VSMCs are preloaded with cholesterol, they exhibit reduced phagocytic and efferocytic capacities compared to macrophage-derived foam cells. It is worth noting that under their experimental conditions, VSMC-derived foam cells expressed the macrophage marker CD68 [181]. Similarly, Clarke et al. showed in vivo that VSMC phagocytosis of apoptotic VSMCs is inhibited by hyperlipidemia [179]. In addition, inefficient apoptotic cell phagocytosis of VSMCs in hyperlipidemic conditions results in secondary necrosis and subsequent release of pro-inflammatory mediators, further aggravating the chronic inflammation associated with atherosclerosis. The inefficient apoptotic cell phagocytosis of VSMCs in hyperlipidemic conditions also results in secondary necrosis and the subsequent release of pro-inflammatory mediators, further aggravating chronic inflammation associated with atherosclerosis [179].

In addition to their inefficient ability to phagocytose apoptotic cells, apoptotic VSMCs significantly contribute to the formation of the necrotic core. Several studies by Clarke et al. showed that VSMC apoptosis alone increases the necrotic core volume of both developing and established plaques in apoE^−/−^ mice [182,183], whereas macrophage apoptosis appears to induce necrotic core growth only in established plaques [184]. Recently, a VSMC-lineage tracing study on apoE-deficient mice fed a high fat diet showed that dedifferentiated Sca1+ VSMCs located near the necrotic core could promote atherosclerosis by evading macrophage efferocytosis [185].

The contribution of VSMC-derived cells to the total necrotic core cells was recently assessed by Chappell and colleagues by the means of a VSMCs-lineage tracing study in mice. Similarly to VSMC-derived cells in the fibrous cap, an average of 70% of the cells in the necrotic core were VSMC-derived. Importantly in the necrotic core, only 10% of VSMC-derived cells expressed αSMA but 50% of them expressed Mac3 [186], meaning that a standard immunostaining procedure would have yielded incorrect numbers of VMSCs and macrophages based on αSMA and Mac3 positivity, respectively. All these studies suggest that VSMCs can promote necrotic core development by at least two distinct mechanisms involving both their reduced efferocytosis capacity and apoptosis.

Senescence is a natural protective mechanism leading to cell division blockade to prevent transmission of defects to progeny cells [187]. However, intimal foam cell senescence mediated by DNA damage, mitochondrial deterioration or oxidative stress is deleterious at all stages of atherosclerosis [188]. In line with this observation, telomere erosion observed in atherosclerotic plaques [189], likely induced by repeated cell divisions of clonal VSMCs, may activate their senescence [190]. Moreover, a recent study from Wang et al. showed that VSMC senescence promotes plaque formation and exhibits marked effects on the fibrous cap and necrotic core. These adverse effects were blunted via reduced DNA damage induced by the expression of telomeric repeat-binding factor-2 (TRF2) in cultured VSMCs. In this study, they also expressed a wild-type form and a loss-of-function mutant (T188A) of TFR2, specifically in VSMCs of apoE^−/−^ mice. TRF2^188A^ transgenic mice showed increased atherosclerosis and necrotic core formation, whereas TRF2 transgenic mice had increased fibrous cap thickness and reduced necrotic core volume [191]. 8oxoG DNA glycosylase I (OGG1) is a base excision repair enzyme which displays anti-atherosclerotic properties [192]. Shah et al. showed that both the necrotic core and the total atherosclerosis area were reduced in ApoE^−/−^ mice expressing a VSMC-restricted OGG1 version compared with ApoE^−/−^ mice deficient in OGG1 [193], suggesting a major role of VSMCs senescence in necrotic core formation irrespective of macrophages.

## 3. “Of Mice and Men”

The first analyses of human atherosclerotic lesions were carried out by Virchow in the 1800s. In his description of the “Vertical section from a sclerotic plate in the aorta in process of fatty degeneration”, Virchow revealed the presence of proliferating cells, VSMCS (“spindle shaped cells”) and foam cells (“fatty degeneration”) at the advanced stages of atherosclerosis [194]. This observation constituted the first description of macrophage-related cells. Almost a century later, Jonasson et al. showed the predominance of macrophages in the different parts of human atherosclerotic plaque by immunohistochemistry, using supposedly “specific markers’’ for leukocytes and for VSMCs (desmin). Macrophages represented 23.9 +/− 3.7% of the fibrous cap and 60.3 +/− 5.1 % of the necrotic core, suggesting their undeniable role in atherosclerosis [195]. To understand the mechanisms of lipid uptake by macrophages in atherosclerosis, Goldstein et al. explored the esterification of cholesterol in different cell types (macrophages or fibroblast) in the presence of extracts from human atherosclerotic plaques. Their results showed that macrophages were the only cell type able to ingest aortic extracts and trigger cholesterol storage/esterification [196]. New technological approaches by mass cytometry (CyTOF) highlighted different clusters and subsets of macrophages such as “resident-like”, ”pro-inflammatory”, “anti-inflammatory”, “involved in foam cell formation” or “involved in cholesterol efflux” [197,198].

While the first animal model of atherosclerosis was developed in rabbits, the use of murine models has enabled the study of the implication of macrophages in atherosclerosis [199]. Models involving bone marrow transplantation used by Linton in 1995 and Herijers in 1997 have introduced the hypothesis that macrophages derived from hematopoietic cells play a crucial role in atherosclerosis. In these pioneering experiments in the field of atherosclerosis, ApoE^−/−^ and LDLR^−/−^ mice were irradiated and respectively transplanted with ApoE^+/+^ and C57BL/6 wild type bone marrow. In the two experiments, results showed a modulation of the lipid profile with a reduction of circulating cholesterol levels [52,200]. Bone marrow transplantation and mouse chimeric models are broadly used to assess the role of macrophages in atherosclerosis. This can range from the study of macrophage pro-inflammatory phenotypes to the study of their metabolic profile. For example, Kennedy et al. highlighted the contribution of CCL3 in the aggravation of atherosclerosis in an LDLR^−/−^ mouse model transplanted with bone marrow from CCL3^−/−^ donors [201]. In another study, Kubo et al. showed the enhancement of atherosclerosis in LDLR^−/−^ mice that received bone marrow from mice with defective Fas-ligand expression (Ipr mice) [202]. Other studies such as those of Nong et al. and Yu et al. have shown the importance of macrophage metabolism in the aggravation of atherosclerosis, notably via hepatic and lipoprotein lipases promoting foam cell formation and the expression of apoE involved in cholesterol efflux and SR-B1 expression [203,204]. Furthermore, the use of murine models allowed the exploration of macrophage subsets in atherosclerosis [205,206].

The involvement of VSMCs in atherosclerosis was established in the 1960s following their identification in both normal and injured arteries of rats, rabbits and humans by electron microscopy [207,208,209]. Most plaque cells including a large part of the abundant foam cells were therefore already thought to be VSMC-derived cells exhibiting an altered phenotype [210]. The subsequent development of immunohistochemistry has provided a clearer indication of the importance of each cell type in early or advanced atherosclerotic plaques. Numerous studies showed that DIT and fibrous cap was mainly constituted of spindle-shaped cells that showed positive results for the anti-SMC monoclonal antibodies (anti-desmin), whereas most of the lipid and necrotic core cells including foam cells were positive for leukocytes antigens such as CD14, T200 and HLA-Class II [195,211,212,213,214]. However, some of these studies pointed at the limits of immunohistochemistry, as many “specific” VSMC markers such as desmin and YPC 1/3.12 are not expressed by the entire VSMC population [195,212]. In addition, several investigations revealed that VSMCs can also express myeloid and lymphoid markers such as CD68 and HLA-DR [75,98,99] leading to probable misinterpretations of histological sections of atherosclerotic plaques in the past. The development of lineage tracing of VSMCs in animal models allowed a precise investigation of VSMC role and fate throughout the different phases of atherosclerosis [97,98,99].

Unlike macrophages that stem from bone marrow, VSMCs are naturally present in the arterial wall, at least in humans. ApoE-deficient mice are the gold-standard model for the investigation of atherosclerosis. Although mouse arteries do not display DIT, which mainly consist of VSMCs in human atherosclerosis prone arteries, a recent study showed that VSMCs contribute to the majority of total foam cells in both WT and apoE^−/−^ VSMC lineage-tracing mice, suggesting that despite structural differences between humans and mouse arteries, VSMCs play a central role in atherosclerosis in both humans and murine models [215]. However, further investigations are needed in order to evaluate the contributions of VSMCs in other in vivo models such as rabbits and rats.

Arterial remodeling after vascular injury such as in atherosclerotic plaques leads to VSMC migration and proliferation towards the intima [216]. In line with this observation, clinical interventions such as angioplasty with stenting may lead to rapid restenosis due to important damage to the arterial endothelium [217]. In 1972, Stemerman and Ross established an in vivo model in order to investigate the arterial remodeling after vascular injury [218]. Arteries from non-human primates were exposed and cannulated with an intravascular balloon catheter to induce endothelial denudation and were then analyzed by electronic microscopy. The migration of VSMCs from the media to the intima as well as their subsequent proliferation were observed within 4 days after injury. After 28 days, the injured intima consisted in multiple layers of VSMCs surrounded by collagen and elastic fibers. Similarities of this in vivo model of vascular injury to the early stages of atherosclerosis strongly supported the role of VSMCs in the development of atherosclerosis [218].

### 3.1. The Limits of Experimentations on Animal Models

Animal experimentation presents a particular interest for the study of atherosclerosis. It allows the investigation of different factors that may influence or directly participate in atherogenesis such as the impact of specific genes or the impact of a diet on a population sharing the same intestinal flora (limitation of the influences of the environment and dietary habits). In addition, it allows ex vivo analysis (for example immunohistology) at early stages of atherosclerosis and also permits investigation of additional atherosclerosis aggravating factors (accelerated atherosclerosis models by proximal middle cerebral artery occlusion, or carotid abrasion) [219,220]. The study by Yin et al. was designed to determine and compare the lipid profiles of different animal models to a common lipid profile of Human dyslipidemia [221].

Among the profiles obtained, that of diabetic rhesus non-human primates was the most similar to the human dyslipidemic profile. However, the use of the murine models remains the rule, due to their rapid reproduction, low cost and the ability to develop some sort of early atheroma [222,223].

However, whereas cholesterol is predominantly transported in LDL particles in humans, the major cholesterol-transporting lipoproteins in mice are high density lipoproteins (HDLs), likely owing to the absence of cholesteryl ester transfer protein (CETP) expression in mice [224]. Beyond this major difference between human and mouse lipoprotein metabolisms, other substantial differences in the development of atherosclerosis can be noted between patients and this animal model. Although in both humans and mice atherosclerosis appears to develop in regions with disturbed blood flow, the primary sites of atheroma formation in mice are the carotid arteries and aorta, but not the coronary arteries. Also, both thick intima and fibrous cap are major features of early atherosclerosis in humans but are not present in mice. Plaque rupture followed by thrombosis is the main cause of heart attack in humans but is very rare in mouse models [225,226]. Calcification is frequently observed in ruptured plaques in humans but is uncommon in mouse models [227].

Despite these significant differences, transgenic mouse models such as ApoE-deficient mice have emerged as a widely used, cheap, convenient and reliable animal model for investigation of human atherosclerosis.

### 3.2. From Formation to Destruction of the Fibrous Cap: The Crucial Role of Macrophages

During the necrotic core expansion, a second structure can be observed on the luminal side of the intima: the fibrous cap. Composed of macrophages, VSMCs and ECM proteins (type I, III, IV and V collagen, elastin), this intimal thickening plays a determinant role in the evolution of atherosclerostic plaques [228,229,230]. In fact, mechanical stress (shear stress), lipid accumulation and endogenous factors (proteases, abundance of macrophages, low number of SMCs, low synthesis of collagen) contributes to the thinning of the fibrous cap and may promote its rupture [231,232].

Macrophage-derived matrix metalloproteinases (MMPs) have long been observed in human vulnerable atherosclerotic plaques and described as a factor promoting the destabilization of the fibrous cap [233,234]. The type of secreted MMPs depends on the macrophage phenotype and environmental factors. For example, MMP-1 (collagenase-1), MMP-3 (stromelysin) and MMP-9 (gelatinase B) can be produced under the influence of ox-LDL, NF-κB activation or in response to growth factors [235,236]. An imbalance between MMP production by macrophages and the levels of their tissue inhibitors (TIMPs) promotes plaque destabilization [237,238]. Furthermore, recent studies showed the implication of macrophage MMP-8 and -9 on VSMC proliferation and differentiation into osteoclasts, suggesting that macrophages would probably orchestrate SMC responses [239,240]. Macrophages and monocytes are described as the major source of protease nexin-1 (PN-1), a serine protease inhibitor (serpin) that may also regulate plaque (de)stabilization [241,242].

A greater concentration of macrophages expressing CD68+, CD11c+ were found in patients suffering from symptomatic acute ischemic attack while asymptomatic patients showed greater amounts of macrophages expressing CD163+ [243]. Analysis by Virmani et al. demonstrated the influence of macrophages in the context of vulnerable plaques. Macrophages respectively constituted 14 +/− 10% and 26 +/− 20% of thin cap fibroatheroma (*n* = 15) and ruptured plaques (*n* = 25) whereas SMC represented 6.6 +/− 10.4% and 0.002 +/− 0.004% [244,245]. Other studies underlined the correlation between macrophage-derived monocytes with the presence of ruptured plaques in patients suffering from coronary artery disease [246,247].

### 3.3. VSMCs Are Indissociable from Fibrous Cap Formation and Rupture

The chronic deposition of LDLs in the subendothelial space results in pathological intimal thickening followed by fibroatheroma characterized by the formation of the fibrous cap surrounding the necrotic core in an attempt of healing, thus promoting plaque stability. A recent multicolored VSMC lineage-tracing study in apoE-deficient mice showed that in the early stages of atherosclerosis, the fibrous cap consists in average of 70% of VSMC-derived cells and that a small but significant percentage (7%) of VSMC-derived cells expressed the macrophage marker mac3 [186]. Interestingly, they also showed that many phenotypes of VSMC-derived cells in atherosclerotic lesions mainly originate from the clonal expansion of only a few medial VSMCs [186]. VSMCs located in the fibrous cap maintain the fibrous cap integrity and thickness by secreting proteoglycans and collagens [233]. In accordance with this hypothesis, Rheker et al. observed that VSMCs colocalize with collagen synthesis in fibrous caps of human arteries [229]. Moreover, early evidence showed that a reduced number of VSMCs resulted in plaque thinning and vulnerability to rupture [248]. More recently, two VSMC-lineage tracing studies highlighted that both KLF4 and octamer-binding transcriptional factor 4 (OCT4), a key player involved in regulating pluripotency in embryonic stem cells, respectively displayed pro- and anti-atherogenic properties due to their effects on VSMC phenotype and fibrous cap thickness. While KLF4 appears to decrease fibrous cap thickness and plaque stability by promoting the phenotypic transition of VSMCs to mesenchymal stem cells and macrophage like pro-inflammatory cells, OCT4 may activate VSMC migration and proliferation from the media to the fibrous cap and therefore contribute to its consolidation [99,249]. Unexpectedly, Gomez et al. showed that inhibition the pro-inflammatory cytokine IL-1β in apoE^−/−^ mice increased plaque fragility by reducing fibrous cap thickness due to increased numbers of macrophages and reduced VSMCs and collagen content. These findings suggest that IL-1β has both detrimental and protective effects in early and late atherosclerosis, respectively [250].

In addition, senescent VSMCs secrete less collagen but release large amounts of MMPs such as MMP-1, MMP2, MMP-3, MMP-8 and MMP-9, therefore contributing to the degradation of the structural matrix that normally strengthens the fibrous cap [240,251,252]. Although the role of VSMCs in the formation of fibrous plaque is undeniable, the mechanisms by which they modulate the fibrous cap thickness and therefore the plaque stability remains to be established.

### 3.4. Intraplaque Hemorrhage and Neovascularization

In the context of advanced atherosclerosis, intraplaque hemorrhage is a key event. The presence of fragile vessels near the necrotic core enhances monocyte and macrophage invasion and promotes red blood cells extravasation [253]. Finn et al. showed that intraplaque hemorrhage macrophages (stimulated by hemoglobin) differ from necrotic core macrophages that tend to accumulate lipids. Indeed, it has been observed that hemoglobin impacts on macrophage differentiation and prevents foam cell formation by downregulating the expression of scavenger receptors [254].

It is well established that intraplaque neovessels originate from adventitial vaso vasorum close to the vascular lesions [255]. Although intraplaque neovessels lack VSMCs [256], recent evidence suggests a role for VSMCs in angiogenesis. It was shown that both mRNA and protein levels of vascular endothelial growth factor-A (VEGF-A) were upregulated in medial SMCs following PPARγ activation by lipid mediators in early human atheromatous aortas [154,257]. Earlier in vitro studies have also shown that VEGF-A can be expressed by SMCs and that endothelial cells cultured on collagen with SMC-conditioned medium became spindle shaped, exhibited an increased proliferative activity and could be organized in capillary-like branching cord structures in collagen gels [258,259]. In addition, it has been shown that SMCs can express angiopoietin-1, a glycoprotein promoting the formation of endothelial tight junctions, thereby reducing endothelial permeability and stabilizing vessels [251]. Intriguingly, VSMCs can also express angiopoietin-2 [245]. In contrast with angiopoietin-1, angiopoietin-2 may be associated with unstable plaque microvessels due to its association with an increased MMP-2 activity [252].

Although further investigations are needed to understand precisely how VSMCs stimulate intraplaque neovascularization, the abovementioned studies demonstrate their angiogenic properties and their pivotal role in the formation of neovessels. Also, whether VSMCs improve or impair neovessel stabilization remains unclear.

### 3.5. Towards Plaque Calcification

Calcification is one of the processes that occurs in advanced plaques. Macrophage cytokines and bone morphogenetic proteins contribute to plaque calcification by promoting osteogenic differentiation of VSMCs [260,261,262]. In addition, the presence of cholesterol crystals associated with calcifications can be a potential marker of plaque vulnerability. Indeed, studies showed that macrophage accumulation was associated with cholesterol crystals and calcifications in atherosclerotic lesions of patients suffering from acute coronary syndrome or myocardial infarction [263,264,265].

VSMCs are closely linked to both medial and intimal calcification [266]. Indeed, VSMCs can differentiate in a number of cell types besides macrophage-like cells, including osteogenic, chondrocytic and osteoclastic cells, resulting in the downregulation of contractile protein expression such as smooth muscle protein-22α (SM22α) and αSMA [267,268]. They may express osteochondrogenic markers including osteopontin, osteocalcin and alkaline phosphatase [266]. Osteogenesis is controlled by many cell regulatory proteins, themselves regulated by physiological and mechanical stimuli. For example, Runx2 and BMP2 are osteogenic markers inducing major changes in VSMC phenotype and subsequent calcification [269,270]. Consistent with this, patients with a rare autosomal recessive disorder caused by a loss of MGP, an inhibitor of BMP2-mediated calcification, display more cartilage and vascular calcification than controls [271]. Among all physiological and mechanical changes promoting VSMC phenotypic switch towards osteogenic, chondrocytic and osteoclastic-like cellular phenotypes, ROS-induced oxidative stress [272,273], VSMC senescence [274], mechanical stress [275,276] and apoptosis [277,278] appear as the most detrimental.

## 4. Promising Macrophage and VSMC Targeting Therapeutic Strategies

To date, most strategies to reduce the risk of atherosclerotic cardiovascular events have been based on lipid lowering treatments in order to limit lipid deposition in artery walls. However, large clinical trials such as FOURIER (Further Cardiovascular Outcomes Research With PCSK9 Inhibition in Subjects with Elevated Risk), have shown that even aggressive LDL-C lowering up to 30 mg/dL with a combination of statins and PCSK9, inhibitors failed to protect 100% of patients from another cardiovascular event [279]. This suggests that other factors independent of LDL-C levels play a role in atherosclerotic plaque development. Macrophage-induced inflammation obviously appears to be the number one suspect responsible for this residual risk. Many therapeutic trials aiming at increasing HDL-C levels have failed to show a significant effect on reducing major cardiovascular events, in spite of an improved reverse-cholesterol transport [280]. Different strategies including the use of pharmacological drugs were designed to raise HDL levels and thus promote macrophage cholesterol efflux. In particular, CETP inhibitors were disappointing for the prevention of cardiovascular events despite increased HDL levels [281]. The notion of HDL functionality was introduced, suggesting that increasing HDL-C alone was not sufficient to prevent atherosclerotic complications. In addition to their function of reverse cholesterol transport, HDLs display anti-inflammatory functions that particularly target macrophages. Injections of reconstituted HDLs or their signature apolipoprotein, apolipoprotein A-I (apoA-I), induced reduction in macrophage content as well as significant decrease of inflammatory M1 and increase in anti-inflammatory M2 macrophage markers in mouse plaques [282,283,284,285]. While apoA-I is best known to activate cholesterol efflux, its beneficial effect on macrophage polarization appears as a good opportunity for further reducing cardiovascular risk in patients. Clinical trials using CER001 have been disappointing in spite of an improvement of the lipid profile [286,287]. The same strategy with different endpoints is under evaluation using CSL112 (CSL Behring), based on the injection of reconstituted HDLs in patients with acute coronary syndrome (AEGIS-II). The results of this study will be available by 2024 (ClinicalTrials.gov Identifier: NCT03473223). Yu and colleagues used nanoparticles containing the synthetic LXR agonist GW3965 (GW) to increase cholesterol efflux from macrophages and hence, promote reverse cholesterol transport from atherosclerotic lesions in LDLR^−/−^ mice. Animals treated with nanoparticles showed a 30% reduction in macrophage content compared with control mice [288]. Similarly, Guo et al. used nanoparticles carrying the LXR agonist, T0901317 (T1317) to increase cholesterol efflux in macrophages in vitro and in APOE-deficient mice. Nanoparticles significantly increased cholesterol efflux from lipid-laden macrophages in vitro and induced plaque regression in mice [289]. As VSMCs also greatly participate in the formation of foam cells, regression of plaques in mice treated with LXR agonists might also be caused by the reduction in VSMC-derived foam cell content. However, VSMC-derived foam cells lack the cellular machinery to properly efflux accumulated cholesterol. For instance VSMCs have reduced expression of ABCA1 [215,290], and low expression of lysosomal acid lipase (LAL) [104] compared with macrophages. As a result, increasing ABCA1 and LAL expression in foam cells (especially in VSMCs) seems to be a promising therapeutic strategy to reduce atherosclerotic plaque burden.

Another macrophage-targeting therapeutic strategy consists in reprogramming macrophages towards an anti-inflammatory phenotype M2. Schistosoma mansoni–derived soluble egg antigens (SEAs) contain a broad range of components that exert immune responses in vitro and in vivo [291]. Wolfs and colleagues showed that in hyperlipidemic LDLR^−/−^ mice, SEA treatment showed a reduction in plaque size of 44% associated with a general anti-inflammatory M2 macrophage phenotype characterized by high IL-10 production and diminished IL-12, NO and TNF-α secretion [292].

The predominant role of VSMCs in atherogenesis (Figure 3) has been clearly shown by recent in vivo studies. They suggest that targeting the transition of VSMCs from a contractile to a synthetic phenotype may be a promising strategy for prevention and reduction of cardiovascular risks. To our knowledge, no study on a potential treatment targeting VMSC phenotypic transition has yet been initiated. However, many single-cell RNA sequencing studies are currently being carried out and will undoubtedly provide valuable information on key genes that might be targeted for the development of innovative VSMCs-targeting strategies [293,294,295,296,297].

## 5. Conclusions

As described in this review, macrophages and VSMCs play key roles in atherosclerosis. Recent studies suggest that the contribution of VSMCs in atherosclerosis is supported by their ability to change phenotype to form foam cells, phagocytes and osteoblasts, or by their ability to build the fibrous cap. These elements are confirmed by older observations highlighting the contribution of VSMCs to the diffuse thickening of the intima in the early stages of atherosclerosis. In this review of the literature, we intended to confront two different points of view of atherogenesis considering that macrophages are the driving force of this pathology whereas VSMCs are passive bystanders and the other way around. The paradigm that puts macrophages in the center stage is changing with the emergence of new technologies such as cell lineage tracing. Most of the evidence obtained for the characterization of atheroma cell composition comes from immunohistology, based on antibodies directed against supposedly specific markers. The plasticity of VSMCs has been underestimated and their capacity to express phagocytic markers common to those expressed by macrophages (CD36, CD68) changed the story. In addition, most of the data accumulated over the last 30 years comes from mouse models, which are very different from humans. First, mice have an inverse lipid profile relative to humans, as they transport cholesterol mostly in HDLs while humans use essentially LDLs. Second, the blood count is also inverted compared to humans: mice mainly have mononuclear leukocytes, while humans have a majority that are polynuclear cells. Since lipids, and particularly LDL-C and leukocytes are the main protagonists of atherogenesis, mouse models may produce enormous bias in the understanding of this pathology. As far as leukocytes are concerned, more and more publications underline the role of polynuclear neutrophils in all stages of atherosclerosis, especially since the discovery of neutrophil extracellular traps [298]. Neutrophils are not the focus of the present review but would deserve more attention as a source of proteases (elastase, cathepsin G, MMP-9...) and pro-oxidant enzymes (myeloperoxidase, NADPH oxidase...). They are by far the first type of leukocytes (65–70%) in humans, while lymphocytes and monocytes are the most represented white cells in mouse blood (>70%). Caution should thus be exercised in the interpretation of data obtained from mouse studies, as the distribution of lipids and leukocytes is completely different from that of humans. The involvement of VSMCs in atherogenesis is increasingly documented and the modulation of their phenotypic change may represent future therapeutic options.

## Figures and Tables

**Figure 1 biomedicines-09-01214-f001:**
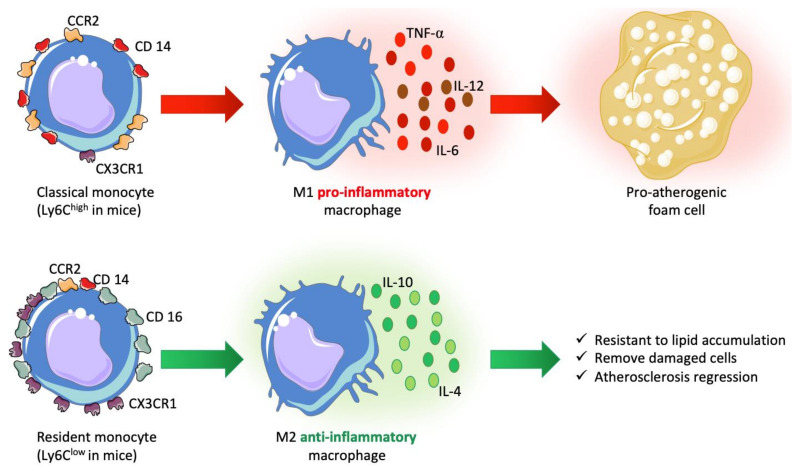
Monocyte subsets and their glycoproteins. Depending on the type of glycoprotein expressed at their membrane, monocytes can be classified in different subsets. Classical monocytes expressing CCR2 and CD14 are associated with pro-inflammatory macrophages. They actively participate in the aggravation of atherosclerosis by secreting pro-inflammatory cytokines and may become foam cells. On the other hand, resident monocytes expressing CX3CR1 and CD16, are associated with anti-inflammatory macrophages, are resistant to lipid accumulation and promote the resolution of atherosclerosis, notably by carrying out an effective efferocytosis.

**Figure 2 biomedicines-09-01214-f002:**
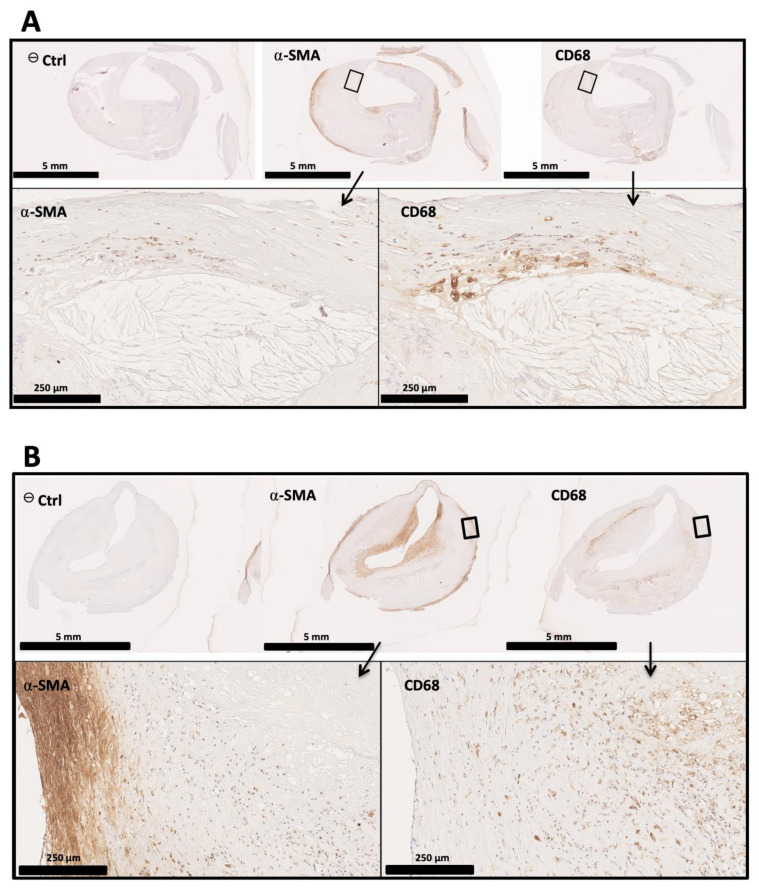
Immunohistological analysis of human atherosclerostic carotid samples. Human atherosclerotic carotid sections were stained for αSMA and CD68. Higher magnifications show areas positive for both staining, containing cells which seem to express both SMA and CD68, suggesting a phenotypic transition of VSMCs from contractile or synthetic towards a phagocytic subset. (**A**) Example of an advanced atherosclerotic section with an important necrotic core and a thin fibrous cap (few αSMA-positive cells). Higher magnifications of the delineated area show the interface between the fibrous cap and the underlying necrotic core containing cholesterol crystals. (**B**) Section of human atheromatous lesion with a thick fibrous cap. Higher magnifications show the interface between the media and the necrotic core, containing cells with potentially intermediate phagocytic/contractile phenotypes.

**Figure 3 biomedicines-09-01214-f003:**
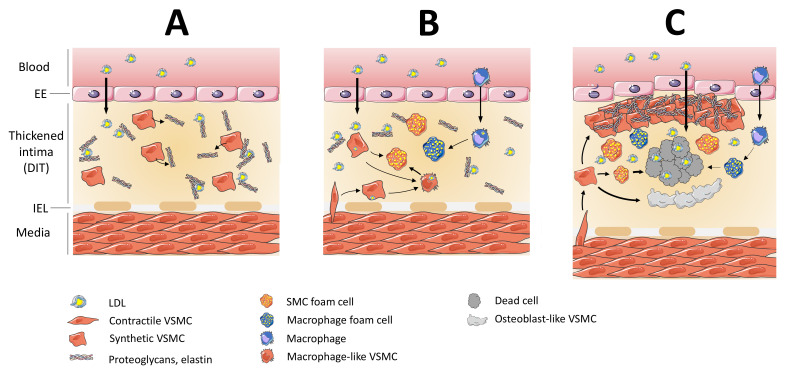
Role of VSMCs in atherosclerosis. (**A**) Pre-atherosclerosis: An excess of circulating LDL particles favors their passage into the sub-endothelial space where they undergo oxidation mediated by surrounding cells and enhanced by the presence of extracellular matrix compounds. This diffuse intimal thickening contains synthetic VSMCs and their secretory products elastin and proteoglycans. VSMCs may also express scavenger receptors, allowing them to ingest oxidized LDLs in a non-regulated fashion and leading to the formation and accumulation of foam cells. (**B**) Macrophage infiltration, VSMC migration and foam cell formation: Monocytes enter the arterial wall and differentiate into macrophages, whereas medial VSMC migrate the intima, where they accumulate LDL particles and become foam cells. (**C**) Fibrous cap and necrotic core formation and calcification: Synthetic VSMCs are responsible for the formation of the fibrous cap, which prevents plaque rupture. However, apoptotic VSMCs also participate in the formation of a necrotic core. Finally, VSMCs may adopt an osteoblast-like cell phenotype, resulting in plaque calcification. EE: Endothelial cells; DIT: Diffuse Intimal Thickening; IEL: Internal elastic lamina.

**Table 1 biomedicines-09-01214-t001:** Main scavenger receptors involved in atherosclerosis.

Cell Type	Receptor	Stimulus	Function	References
**Macrophages**	Scavenger receptor-A1 (SR-A1)	Visfatin, ox-LDL	Ox-LDL uptake, cell apoptosis, foam cell formation	[127,128,129]
	Lectin-like oxidized LDL receptor-1 (LOX-1)	Ox-LDL, ER stress, modified lipoproteins, advanced glycation end-products (AGEs)	Ox-LDL uptake, inhibition of macrophage migration, increase in foam cell formation	[130,131,132,133]
	CD 36	Ox-LDL, ox-phospholipids, IL-34, visfatin	Ox-LDL uptake, cytokine release, NLRP3 inflammasome activation, increased in foam cell formation	[130,131,134,135]
	CD 68	Ox-LDL	Ox-LDL uptake, contribution to inflammation	[136,137]
	CD 146	Ox-LDL	Foam cell formation, macrophage retention	[138]
	TLR4	Ox-LDL, LPS, saturated fatty acids	Mediates inflammatory response	[134,135,139]
	Macrophage scavenger receptor -1 (MSR-1)	Native or modified LDL (acetylated, oxidized	Ox-LDL uptake	[140,141]
**SMC**	Scavenger receptor-A1 (SR-A1)	Phorbol esters, ROS, ox-LDL	Ox-LDL uptake, foam cell formation	[142,143]
	CD 36	Free fatty acids (Oleic acid), ox-LDL	Free fatty acid and ox-LDL uptake, Foam cell formation	[144]
	Scavenger Receptor BI/II	Intracellular cholesterol content	Cholesterol ester uptake, apoB-containing lipoproteins	[145,146]
	Lectin-like oxidized LDL receptor-1 (LOX-1)	LPS, TNF-a, IL-1β, IFN-y, ox-LDL, shear stress	SMC proliferation, ox-LDL uptake, foam cell formation	[131,147]

## Data Availability

Not Applicable.

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
