# Peer review of "Macrophages in Atherosclerosis, First or Second Row Players?"

_biomedicines, 2021, doi:10.3390/biomedicines9091214_

Round 1

Reviewer 1 Report

The revision is acceptable.

Author Response

We thank the Reviewer #1 for considering our manuscript acceptable for publication. 

Reviewer 2 Report

Eloise Checkouri et al. provided an improved manuscript and have answered meticulously to the majority of reviewer’s comments. Last clarifications will be very helpful if provided before final acceptance.

  • Point n4 is still not clear, especially the part regarding the “negative feedback” in the presence of LPS.
  • About point n6: the reference below is also much more detailed about the role of Ly6Chigh and Ly6Clow in atherosclerosis and role of Treml4.

Gonzalez-Cotto M, Guo L, Karwan M, Sen SK, Barb J, Collado CJ, Elloumi F, Palmieri EM, Boelte K, Kolodgie FD, Finn AV, Biesecker LG and McVicar DW (2020) TREML4 Promotes Inflammatory Programs in Human and Murine Macrophages and Alters Atherosclerosis Lesion Composition in the Apolipoprotein E Deficient Mouse. Front. Immunol. 11:397. doi: 10.3389/fimmu.2020.00397

  • Information about metabolic characteristics of “M1” and “M2” macrophages still need to be enriched (revision point n13; paragraph 2.1). Please refer to work done in immunometabolism.
  • Figure 2 does not need a letter A if there is no B (no B can be seen in the figure even though the authors refer to it in the legend).

Author Response

Eloise Checkouri et al. provided an improved manuscript and have answered meticulously to the majority of reviewer’s comments. Last clarifications will be very helpful if provided before final acceptance.

Point n4 is still not clear, especially the part regarding the “negative feedback” in the presence of LPS.

About point n6: the reference below is also much more detailed about the role of Ly6Chigh and Ly6Clow in atherosclerosis and role of Treml4.

Gonzalez-Cotto M, Guo L, Karwan M, Sen SK, Barb J, Collado CJ, Elloumi F, Palmieri EM, Boelte K, Kolodgie FD, Finn AV, Biesecker LG and McVicar DW (2020) TREML4 Promotes Inflammatory Programs in Human and Murine Macrophages and Alters Atherosclerosis Lesion Composition in the Apolipoprotein E Deficient Mouse. Front. Immunol. 11:397. doi: 10.3389/fimmu.2020.00397

The points 4 and 6 have been clarified and expanded. The suggested reference has been added.

Information about metabolic characteristics of “M1” and “M2” macrophages still need to be enriched (revision point n13; paragraph 2.1). Please refer to work done in immunometabolism.

This point has also been expanded (in red in the new version of the manuscript)

Figure 2 does not need a letter A if there is no B (no B can be seen in the figure even though the authors refer to it in the legend).

We thank the Reviewer #2 for this remark. The part B of the figure was missing; it has been added in the new version of the manuscript.

This manuscript is a resubmission of an earlier submission. The following is a list of the peer review reports and author responses from that submission.

Round 1

Reviewer 1 Report

The manuscript by Checkouri et al provides a great review of the literature and the role of macrophages and vascular smooth muscle cells in atherosclerosis. The authors thoroughly describe the role of each cell type.

Can the authors provide information on current therapeutics targeting these cells or current/past clinical trials that targeted these cells?

Reviewer 2 Report

In this review article, Eloise Checkouri et al introduce the scientific need of summarizing the current state of understanding on atherosclerosis and how to discriminate the role that macrophages and SMC play in the different states of the disease.

After descriptions of the mechanisms for foam cell formation and early stages of disease, authors describe major differences in important aspects of atherosclerosis in human vs mouse and focus on the late stages of disease.

The potential for this review article is very interesting, knowing the need in the literature for more clarifications of the mechanisms of atherosclerosis and the cell types that play central roles in its pathogenesis. However, authors do not provide sufficient evidence for fair comparisons between macrophages and SMC and never discuss and propose future directions in the field. I believe the current manuscript is not suitable for publication in this journal.

Major points:

  1. Too many questions should be avoided in the abstract for a more organic soundness.
  2. In many points throughout the manuscript, authors make not so much scientific sounding comment that could be omitted.
  3. In paragraph 1.1 more details need to be added and actual research papers should be cited, not only reviews.
  4. Authors should explain the mechanisms and cite literature about why mod-LDL are not endocytosed via the regular pathway.
  5. In paragraph 1.2, authors should consider adding data correlating numbers/markers of monocyte/macrophages with the severity of the plaques.
  6. While discussing roles of monocytes Ly6Chigh and Ly6Clow and progression of atherosclerosis, they could cite recent work on involvement of these populations expressing the marker Treml4 and diseases progression.
  7. It would be important to draw major conclusions about CCLs and CCRs. Are there definite data on amelioration/aggravation of atherosclerosis by manipulating these?
  8. It would be helpful to explain how the medial layer of arteries is composed. On this point, authors could provide a detailed figure/drawing/schematic of the vessel of artery with endothelial cells, lipids, macrophages and SMCs.
  9. The ability of VSMC to become foamy needs to be introduced earlier in the manuscript as it is mentioned abruptly at the beginning of paragraph 2 but only later on is better explained.
  10. It would be helpful to describe the mechanism by which lipoprotein enter the DIT. On this point, it is not clear how SMCs are accessible within the DIT. DIT itself needs to be defined better. Is it just a layer of extracellular matrix and proteins outside the cells?
  11. It would be important to describe what synthetic SMCs and EMC are. This concept is brought up many times in the manuscript but a good definition is lacking.
  12. Authors need to explain the mechanisms by which during early atherosclerosis the basal laminae is degraded.
  13. From paragraph 2.1 on, many parts need to be re-written. Information on macrophages is very poor. It needs to be well defined in terms of markers, cytokines, gene expression and metabolism in macrophages in early vs late plaques.

Later on indeed, information about macrophage-derived promotion of angiogenesis is completely missing. If authors want to advocate for SMCs, they also need to be fair and add all the evidence for macrophage roles as well.

  1. It would be important to give clear details and references on what markers/properties SMCs may share with macrophages and when during pathogenesis and progression of disease.
  2. During ox-LDL metabolism, it’s not clear if free cholesterol and fatty acids are released in the cytosol or extracellularly, and where ACAT1 is located.
  3. It would be helpful to make a figure/drawing of cholesterol-induced apoptosis.
  4. Authors should specify if SMCs are involved at all with platelets uptake (erythrocyte phagocytosis). On this same note, authors should provide more references on SMC being able to perform phagocytosis/efferocytosis.
  5. Being a review, data in Figure 2-3 could be just cited in the text without the need for a figure. Authors could just discuss the need in the field to evaluate non-macrophage markers in the lesions to really dissect the role of macrophages vs SMC.
  6. Authors should discuss any information in the literature about “senescence” detected in macrophages at any time during disease.
  7. Authors should better explain why animal studies investigating the role of VSMCs require a previous vascular injury-induced lesion.
  8. Structural differences between humans and mouse arteries need to be described in detail.
  9. It would be helpful to add more details from ref 223,224 and mechanisms at the basis of macrophage-mediated orchestration of SMC responses towards osteoclast differentiation. On the same note authors should add information about macrophage ability to differentiate in osteogenic cells too in this situation.
  10. Authors should add more details about the study in ref 232 and about techniques of SMC lineage tracing.
  11. It is not clear if inhibition of cytokines like IL1b is beneficial in atherosclerosis and to what extent, especially in relation with the status of the fibrous cap.
  12. Authors should clarify if VSMCs only secrete MMP9 or other MMPs as well (like macrophages).
  13. In the conclusions and throughout the text there is no speculations by the authors, or any suggestion on how the field can progress. Given the authors’ biased point in favor of major role of SMCs for many stages of atherosclerosis, they should provide and discuss ideas and methods for the field to move forward in the right direction of understanding and dissecting the role ascribed to each cell type in each stages of the disease, and how better mouse work can be translated to human or viceversa how human disease can find appropriated means to be studied in mice.

Minor:

  1. Reference should be added to the statement that “the endothelium is therefore more permeable to low-density lipoproteins”.
  2. It is not clear in which cells selectins and their ligands are expressed.
  3. Please define PIT the first time it is mentioned in the text.
  4. It would be important to add more details about ref 168 and how TRF2 epxpression is induced in ApoE-/- .
  5. Please clarify what "anti-macrophage" monoclonal antibody was used for assessing the lipid and necrotic core cells including foam cells (lines 467-469).
  6. Please define percutaneous angioplasty.
  7. Authors should add appropriate references to the statements of different predominant circulating forms of cholesterol in humans and mice as well as for the absence of cholesteryl ester transfer protein (CETP) in rodents.
  8. Drawing in Fig 4A-B should be the same size.